# Multivariate Assessment of Low-Flow Hazards via Copulas: The Case Study of the Çoruh Basin (Turkey)

**Fatih Tosunoğlu [1], Gianfausto Salvadori [2],\***  **and Muhammet Yilmaz [1]**

[1]   Department of Civil Engineering, Erzurum Technical University, 25050 Erzurum, Turkey;
     ftosunoglu@erzurum.edu.tr (F.T.); muhammet.yilmaz@erzurum.edu.tr (M.Y.)
[2]   Dipartimento di Matematica e Fisica, Università del Salento, I-73100 Lecce, Italy
\*   Correspondence: gianfausto.salvadori@unisalento.it

**Abstract:** Bivariate modeling and hazard assessment of low flows are performed exploiting copulas. 7-day low flows observed, respectively, in the upper, middle and lower parts of the Çoruh basin (Turkey) are examined, considering three pairs of certified stations located in different sub-basins. A thorough statistical analysis indicates that the GEV distribution can be used to model the marginal behavior of the low-flow. The joint distributions at each part are modeled via a dozen of copula families. As a result, the Husler–Reiss copula adequately fits the joint low flows in the upper part, while the t-Student copula turns out to best fit the other parts. In order to assess the low-flow hazard, these copulas are then used to compute joint return periods and failure probabilities under a critical bivariate "AND" hazard scenario. The results indicate that the middle and lower parts of the Çoruh basin are likely to experience the largest drought hazards. As a novelty, the statistical tools used allow to objectively quantify drought threatening in a thorough multivariate perspective, which involves distributional analysis, frequency analysis (return periods) and hazard analysis (failure probabilities).

**Keywords:** low-flow; hazard assessment; return period; failure probability; copula; Çoruh basin

## 1. Introduction

In recent years, the increase of water demand because of population growth, technological progress and water pollution has caused various types of water resource problems: therefore, a more efficient and effective management has gained the attention of researchers. Droughts, a natural and recurrent phenomenon all around the world, have a direct impact on natural ecosystems and human activities such as agriculture, tourism, industry, energy and transport, and may affect large areas for long times [1]. Different drought definitions are proposed in literature, and droughts are generally classified in four types: meteorological, hydrological, agricultural and socio-economic [2]. Hydrological droughts (entailing a significant reduction of available surface and subsurface water variables, such as river streamflow, groundwater, reservoir and lake levels), are considered as having the most serious impacts on water supply [3,4]. In particular, the streamflows are considered as important indicators that define the status of surface water resources [5]. Indeed, a hydrological drought is directly related to the reduction of river streamflows with respect to normal conditions [6].

The low-flow index is an important indicator in the analysis of river streamflows. Estimation of low-flows is important for hydrological and water resource management, as it is a critical factor for basin restoration, water quality regulation, reservoir storage design, hydro-electric power generation in arid periods and habitat protection [7,8]. There is a variety of measures and indices used to characterize low-flow, including mean annual runoff, mean daily flow, median flow and absolute minimum flow. Among these, the *7-day annual minimum flow* (AM7) is a widely used low-flow indicator, since it is important for regulation of environmental flows and qualitative planning of river streamflows [9].

Frequency analysis of AM7 series is vital for understanding the behavior of hydrologic variables and for an accurate hazard assessment in engineering problems [10]. Hence, a number of studies have been published on the univariate frequency analysis of low-flows (i.e., [11–13]. However, the rivers are composed of several branches, and the analysis of the dependence between these branches is fundamental for water resource management. In turn, for a comprehensive and accurate hazard assessment of low-flow events, an univariate frequency analysis may be insufficient and/or inaccurate to account for all the relevant probabilistic and statistical structures of interest. Therefore, joint modeling of the variables of interest is required for the analysis of these events. In recent years, as a new multivariate approach, *copulas* have been used in several hydrological fields—see below. Copulas do not have the limitations of traditional multivariate distributions, such as Normal, Lognormal, Exponential, Gamma and Extreme Value, and can model the dependence structure of any set of random variables. In addition, copulas can easily join complex marginal distributions in the models of interest [14].

In literature, copulas have been widely adopted for the multivariate frequency analysis of hydrological processes (i.e., [1,15–24]. However, only a limited number of studies concerning low-flow events are available. For instance, ref. [25] employed copulas to analyze the joint distribution of 7-day low and high flows at two hydrological stations located in West River and North River, respectively, in Pearl River Basin, China. The results of this study indicate that the probability of simultaneous occurrence of low and high flows for these stations is very small. Hence, they emphasized that reductions in the loss of life and property due to extreme hydrological events can be observed. Ref. [26] presented a copula-based low-flow frequency framework for evaluating drought risk in Dez River Basin, Iran. Their results reveal that while the Gumbel-Hougaard copula was suitable for joint frequency analysis at the upstream stations, the Frank copula had the best agreement with the corresponding empirical one at downstream stations. Furthermore, they concluded that severe droughts are more likely to appear at the upstream of the basin as compared to its downstream.

In the light of this short literature review, it is clear that the copula based low-flow modeling is still in its infancy, and deserves more attention. The main purpose of this study is to develop a copula-based bivariate joint modelling of 7-day low-flows in different sub-basins of the Çoruh Basin (Turkey). For this purpose: (1) the marginal probability distributions of the 7-day low-flows at each part of the basin are computed; (2) the joint distributions of the low-flows are modeled using suitable bivariate copula families; (3) joint return periods and failure probabilities of the low-flows are derived exploiting the best performing copulas for each part, and a bivariate hazard assessment is carried out. To the best knowledge of the authors, this is the first study that (i) applies copulas to the joint modeling of low-flows in order to assess the multivariate hazard via hazard scenarios and failure probabilities, and (ii) investigates the natural behavior of a basin located in Turkey.

## 2. Material and Methods

### 2.1. Study Area and Data

In this study, the Çoruh river basin, located in eastern Black Sea region in Turkey (see Figure 1), was selected as the area of interest. The Çoruh river originates from Mescit Mountains in the Bayburt Province and flows through the East Anatolia into the Black Sea near Batumi City in Georgia, after its main course of 431 km [27]. The total drainage area of the basin is about 21,000 km$^2$, and 91% of this area is located in Turkey, and the remaining is in Georgia [28]. The annual average rainfall is approximately 480 mm and the annual mean flow of the Çoruh river is 6.3 billion m$^3$ [29]. The Çoruh basin consists of three sub-basins (namely upper, middle and lower), located between the Black Sea and Eastern Anatolia regions of Turkey. Thus, the characteristics of both the Black Sea and the continental climate type are experienced in the Çoruh basin. In the upper basin, generally, winter seasons are snow-cold and summer seasons are dry-hot. In the lower basin, all seasons are rainy, and humidity

effects are strongly felt due to the Black Sea climate. The middle basin has a transitional climate between the previous two sub-basins.

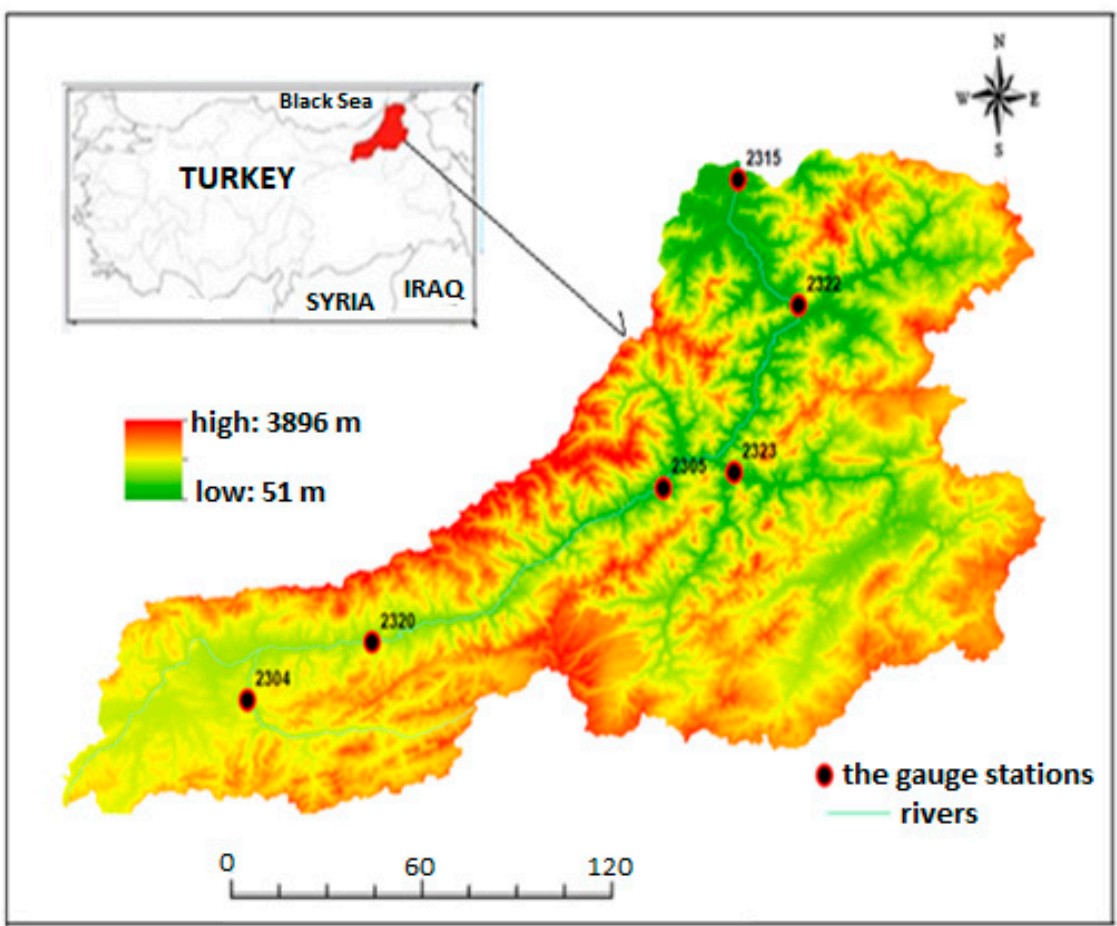

**Figure 1.** Study area and spatial distribution of the gauge stations used in the present study.

Figure 1 shows the map of the study area and the locations of the six certified gauge sites used in this study: the pairs of stations 2304–2320, 2305–2323 and 2322–2315 are located in the upper, middle and lower Çoruh basins, respectively. The Government of Turkey has planned a large scale development project in the Çoruh river basin. The goal is to ensure Turkey's energy needs via renewable sources. As a result, a lot of large-scale water structures, hydroelectric power plants and regulators were built in the basin, and others are under construction or in a planning stage. For example, Deriner dam located in Artvin Province in the lower Çoruh basin is the third highest dam in the world with a height of 249 m. It has a storage volume of 1.97 billion m$^3$ and is considered the second largest reservoir in the Çoruh basin. The Yusufeli dam, presently under construction, has the highest storage capacity among the dams operating on the Çoruh River, and is expected to generate 1.888 GWh/year of electricity [30]. For the reasons mentioned above, the Çoruh basin was selected as a study area for researchers interested in topics such as flow, rainfall, drought and flood analyses (i.e., [28,31–34]).

In this study, daily streamflow records from six certified gauge stations located in the Çoruh basin are used for multivariate frequency analysis of 7-day low-flow (AM7), which is the lowest average flow of seven consecutive days within 1 year (see Figure 2). The reason for choosing these stations is that the observations have not been influenced by human intervention, showing homogeneity and no missing values: this may represent an exceptional case study concerning natural streamflow dynamics; possibly, one of the last available in the world. The stations also have the longest temporal lengths

in the basin, suitable for statistical applications and are located in crucial sites which well reflect streamflow characteristics of low-, middle- and upper-parts of the Çoruh basin.

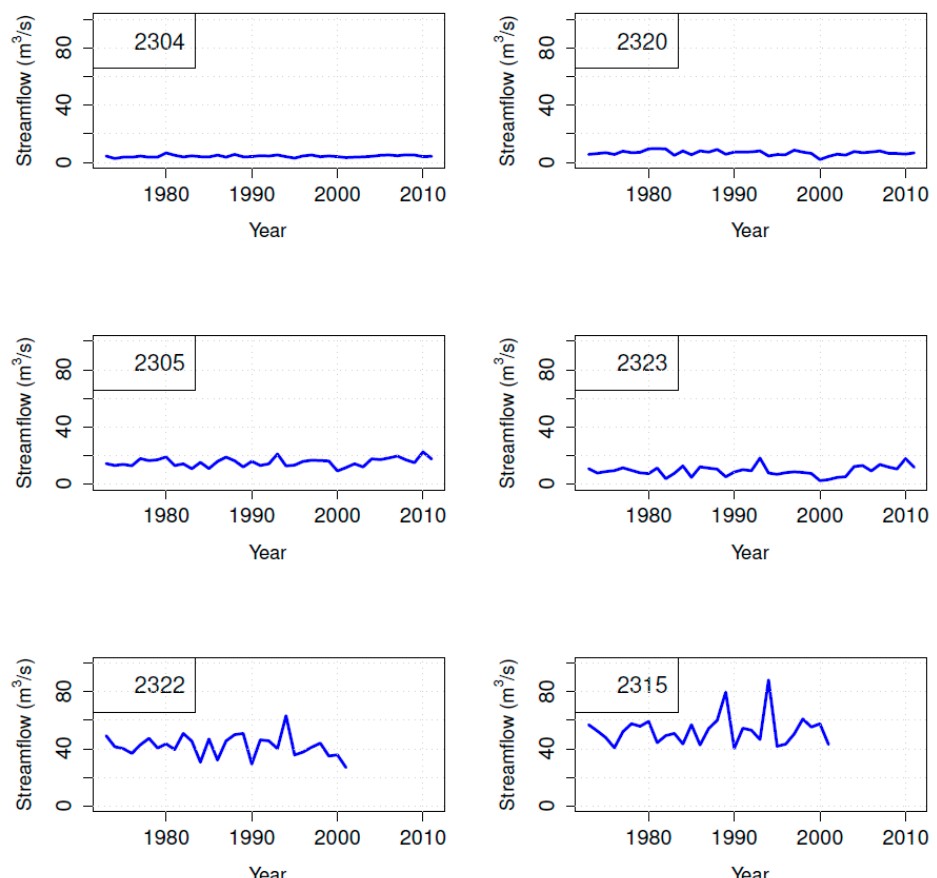

**Figure 2.** Time series of the 7-day low-flow data series used: the labels indicate the station's codes.

Table 1 shows some basic information about the stations and important statistical features of the data. Overall, the mean value of the AM7 series varies between about 4.32 and 53.09 m³/s (stations 2304 and 2315, respectively). The standard deviation of the daily streamflow varies approximately between 0.76 and 10.66 m³/s, and its maximum value is observed at station 2315. The skewness coefficient shows the degree of asymmetry around the mean: it ranges from −0.38 to 1.60, implying that the data have a positive skewness (except for one station). The coefficient of variation is the ratio of the standard deviation to the average of the observed data series and is a dimensionless value: it ranges from 0.18 to 0.37.

**Table 1.** Basic statistics of the gauge stations for 7-day low-flow series (AM7).

| Basin | Station Code | Observation Period | Mean (m³/s) | Standard Deviation (m³/s) | Coefficient of Skewness | Coefficient of Variation |
|---|---|---|---|---|---|---|
| Upper Çoruh | 2304 | 1973–2011 | 4.32 | 0.76 | 0.57 | 0.18 |
| | 2320 | 1973–2011 | 6.70 | 1.57 | −0.38 | 0.23 |
| Middle Çoruh | 2305 | 1973–2011 | 15.43 | 2.93 | 0.19 | 0.19 |
| | 2323 | 1973–2011 | 9.32 | 3.49 | 0.36 | 0.37 |
| Lower Çoruh | 2322 | 1972–2000 | 41.91 | 7.63 | 0.32 | 0.18 |
| | 2315 | 1972–2000 | 53.09 | 10.66 | 1.60 | 0.20 |

Figure 3 provides full information about the distributional features of the data. On the one hand (top panel), the boxplots of the actual observations give the possibility to better and thoroughly

appreciate/evaluate the differences of the distributions of the streamflows recorded in the six stations. On the other hand (bottom panel), suitable standardized indices (computed by normalizing the data, i.e., by subtracting the mean and dividing by the standard deviation) give the possibility to compare the standardized flows, which turn out to be rather homogeneous from a distributional point of view, and give an idea of the regularity of the stream; actually, as we shall show below, it will not be a surprise that the generalized extreme value (GEV) distribution provides valuable univariate fits at all the six stations. In Figure 3, we objectively quantify the differences and the similarities between the different parts of the Çoruh basin; essentially, different sub-basins experience different streamflow dynamics (as obviously expected), but (as a novel distributional information) a homogenous standardized behavior is present, which represents an important guidance from a hydrological point of view.

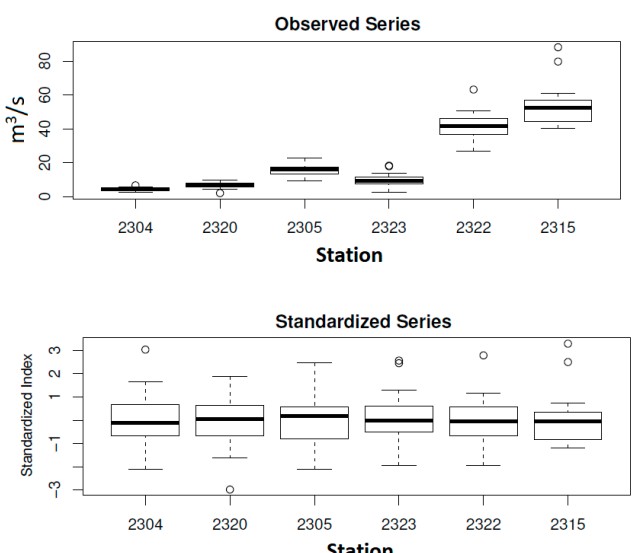

**Figure 3.** Box-plots showing the distributional features of the observed time series (**top panel**), and of the standardized time series (**bottom panel**)—see text.

## 2.2. Marginal Distributions of Low-Flow Time Series

In the conventional frequency analysis, the data are assumed to be realizations of independent and identically distributed (IID) random variables. In order to check the IID features, the Mann–Kendall trend test (see Table 2) and the serial autocorrelation graphs (see Figure 4) are used. As a result, at a 5% level, the series can be considered as stationary ones. In addition, at a 5% level, the autocorrelation plots shown in Figure 4 indicate that the independence assumption cannot be rejected. Actually, these results are also supported by the outcomes of the Box–Pierce and the Ljung–Box tests for checking serial dependence (not shown), since the corresponding $p$-values are larger than 5%. Furthermore, additional suitable statistical Change-Point tests [35], indicating whether the probability distributions of the data might have changed with time, show that the stationarity assumption cannot be rejected.

**Table 2.** The Mann–Kendall test for 7-day low-flow series.

| Station Code | Computed $z$-Values | $p$-Values |
| --- | --- | --- |
| 2304 | 1.839 | 0.066 |
| 2320 | −0.907 | 0.364 |
| 2305 | 1.754 | 0.079 |
| 2323 | 0.653 | 0.514 |
| 2322 | −1.201 | 0.230 |
| 2315 | 0.094 | 0.925 |

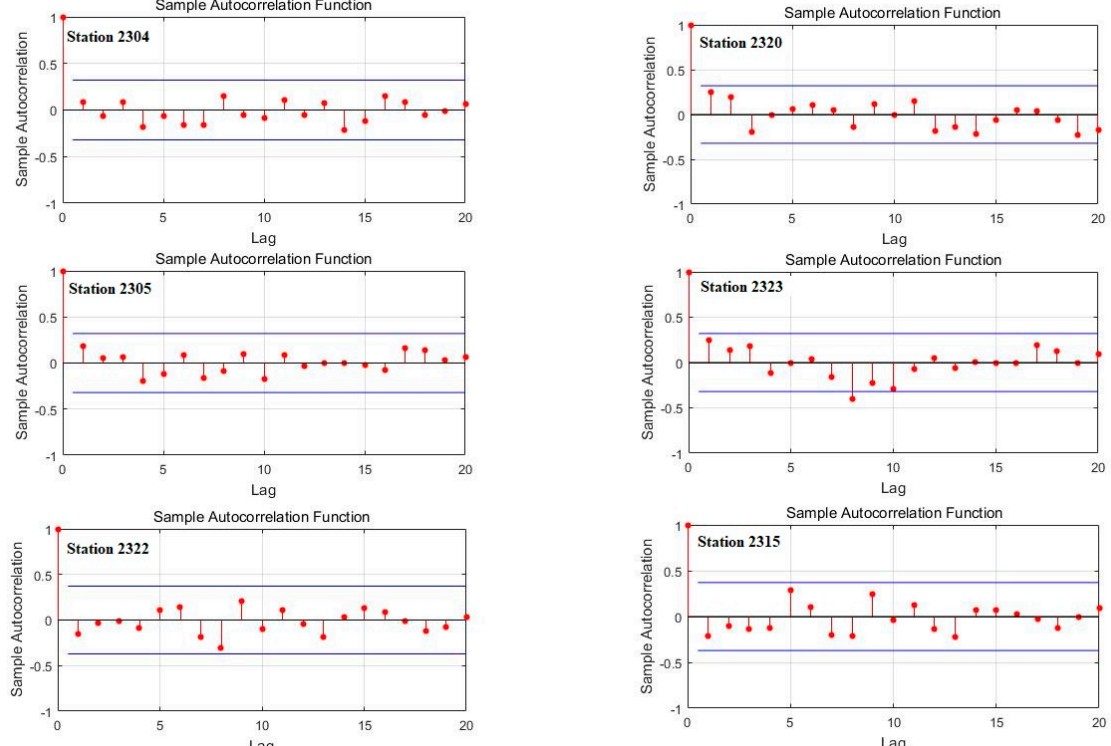

**Figure 4.** Autocorrelation plots of the 7-day low-flow series.

Prior to a multivariate frequency analysis of low-flow events, suitable univariate distributions must firstly be determined. However, there is no accepted common probability distributions for low-flow events in literature [11]. In hydrology, many previous studies proposed that low-flow data could be modeled by different forms of Weibull, Gumbel, Pearson Type III and Log-normal distributions [36]. Here, eleven widely used distributions (viz., Gamma, Gumbel, GEV, Generalized Pareto Distribution, Logistic, Log-Logistic, Normal, two- and three-parameter Lognormal, Pearson Type-III and Weibull) are considered as candidate marginals for low-flow data. The standard Maximum Likelihood procedure is used to fit the parameters.

### 2.3. Copulas of Low-Flow Time Series

In 1959, Sklar [37] proved a fundamental Representation Theorem that expresses the joint probability distribution of any set of random variables in terms of a suitable copula (i.e., a dependence structure) and the corresponding univariate marginals. Since hydrological events are generally multidimensional (as in the present case—see below), the joint modeling of the random dynamics of several random variables is necessary [19,38]. The most important advantage of using copulas is that it is possible to choose the marginals independently of the dependence structure.

In the present work, various families of copulas are used for the bivariate modeling of the 7-day low-flow series observed at the pairs of gauge stations mentioned above:

- Archimedean (Clayton, Frank, Gumbel–Hougaard, Joe);
- Elliptical (Student-t and Gaussian);
- Extreme Value (Galambos, Husler–Reiss, Tawn);
- Special (Farlie–Gumbel–Morgenstern, Plackett)

The corresponding analytical expressions and specific properties can be found in [39] and [14]. The standard Maximum Likelihood procedure is used to fit the parameters. The selection of a "best" copula model is as follows. First, a Cramér-von Mises (CvM) Goodness-of-Fit (GoF) test is carried out,

to check whether a given copula should be rejected or not (whether it is admissible, and statistically compatible, with the available data): A CvM approach at a standard 5% critical level, instead of a Kolmogorov-Smirnov (KS) one, is used here, since the former turns out to be more robust and powerful than the KS one in the multivariate case [40]. Secondly, considering only the admissible copulas (i.e., those that passed the GoF test), the corrected Akaike Information Criterion (AIC) is used to select the best one—see below.

The first step in any copula analysis consists of assessing the degree of association of the variables of interest. A traditional measure of association is represented by the Kendall's τ rank correlation coefficient [14,39], a non-parametric measure of concordance/discordance that does not suffer from many of the limitations of the more commonly used Pearson's ρ linear correlation coefficient. The estimates of both coefficients will be presented in Section 3.2.

### 2.4. Bivariate Return Periods

Joint probabilities of low-flow characteristics can provide useful information for an efficient management and planning of water resources systems: these can be computed in terms of copulas, since copulas are necessary and sufficient to calculate the probabilities of interest, as shown in [41]. In particular, here a bivariate analysis is mandatory, in order to correctly assess the occurrence of critical events of dependent random variables.

The notion of return period (RP) is commonly used in hydrology for quantifying the dangerousness of specific occurrences. A general definition of the RP $T$, which works both in the univariate and in the multivariate case, is as follows [18,41]

$$T = \frac{\mu}{P(X \in HS)} \tag{1}$$

where $\mu$ is the average inter-arrival time between the occurrences in the time series considered (here $\mu = 1$ year, since we use annual data), $X$ is the variable of interest (i.e., discharge) and $HS$ is a so-called hazard scenario, i.e., a region containing the occurrences reputed to be dangerous according to suitable criteria—see below.

In the following, we compute the RPs of interest under the AND paradigm, the one accounting for the most dangerous situations—see [42]. In the AND approach it is *necessary* that all the variables are smaller than the corresponding thresholds to experience a threatening condition: the AND $HS$ is defined as $\{X \leq x \text{ AND } Y \leq y\}$. Obviously, the traditional univariate RP hazard scenarios $\{X \leq x\}$ and $\{Y \leq y\}$ include the AND one: in turn, the corresponding univariate RPs are smaller than the bivariate AND ones [18]. However, it is important to stress that the comparison of different RPs is a delicate issue [42,43], since the failure mechanisms generating adverse conditions (i.e., the hazard scenarios at play) are different, and the results should not be mis-interpreted. The formula for the low-flow joint AND analysis is as follows:

$$T_{\text{AND}} = \frac{1}{P(X \leq x \text{ AND } Y \leq y)} = \frac{1}{C(F_X(x), F_Y(y))} \tag{2}$$

where $C$ is the copula of the pair $(X, Y)$. The computation of the return periods of interest will be presented in Section 3.3.

### 2.5. Bivariate Failure Probabilities

The notion of failure probability (FP) may provide a consistent way of assessing the hydrological (and, more generally, the environmental) hazard—see [42] and references therein. Despite its name, a failure probability is not the probability of a "structural" collapse: simply, it is the probability of observing an occurrence of the phenomenon of interest in a prescribed hazard scenario at least once in a given design life time; clearly, this represents a threatening situation. In the present annual framework, the FP $H$ can be computed as in [42]:

$$H = 1 - (1 - p)^n \tag{3}$$

where $p$ is the probability of the hazard scenario considered, and $n$ is the design life time (in years)—see below.

## 3. Results

### 3.1. Univariate Distributions of 7-Day Low-Flow Series

As already mentioned above (see Figure 3), the GEV distribution turns out to adequately fit the time series at all stations: in fact, in all cases, the Monte Carlo $p$-values of the Kolmogorov–Smirnov (KS) GoF test are always larger than the standard 5% critical level, as shown in Figures 5–7.

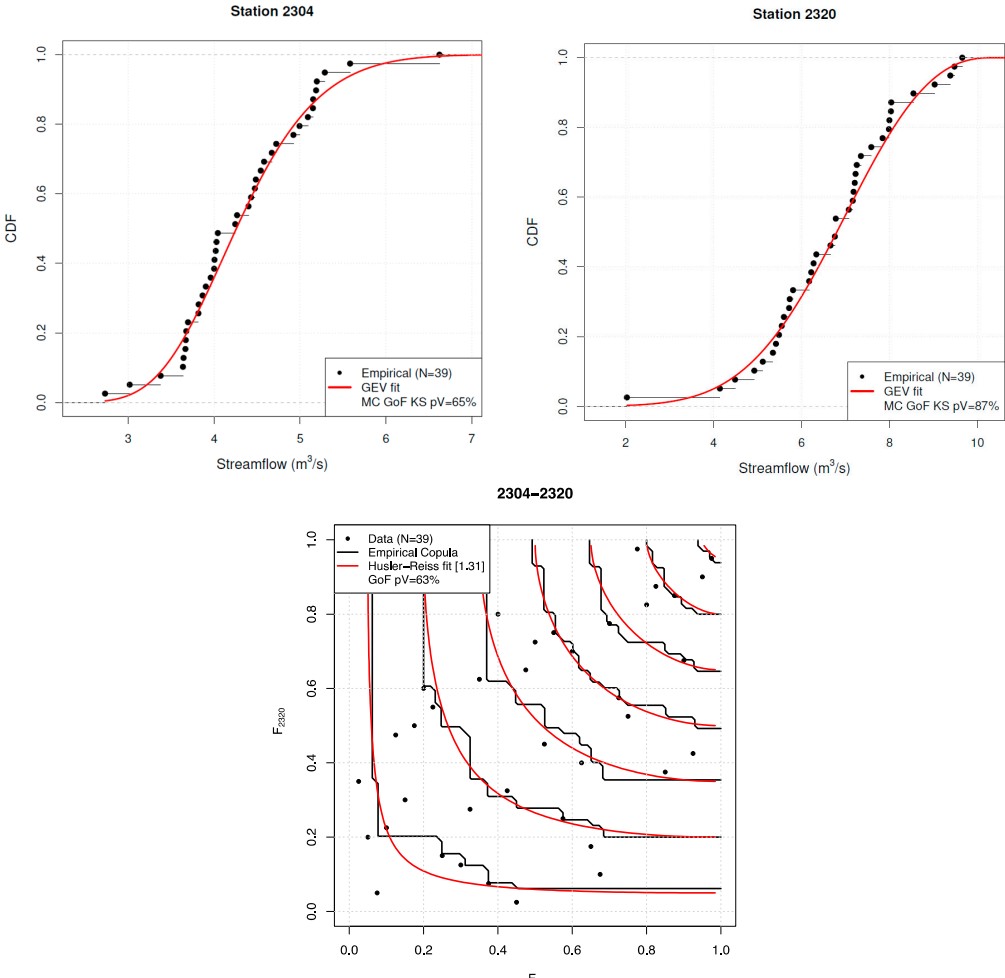

**Figure 5.** Upper Çoruh basin. (**Top row**) Generalized extreme value (GEV) fits of the 7-day low flow series in the upper Çoruh basin. Shown: the empirical CDF (black markers), the fitted CDF (red line), the sample size N and an approximate Monte Carlo Goodness-of-Fit test $p$-value of Kolmogorov–Smirnov type. (**Bottom row**) Copula fit. Shown: the pseudo-observations (markers), the empirical copula (black lines) and the fitted copula family (red lines). Additionally shown are the sample size N, an MLE estimate of the copula parameter, and an approximate Monte Carlo Goodness-of-Fit test $p$-value of Cramér-von Mises type.

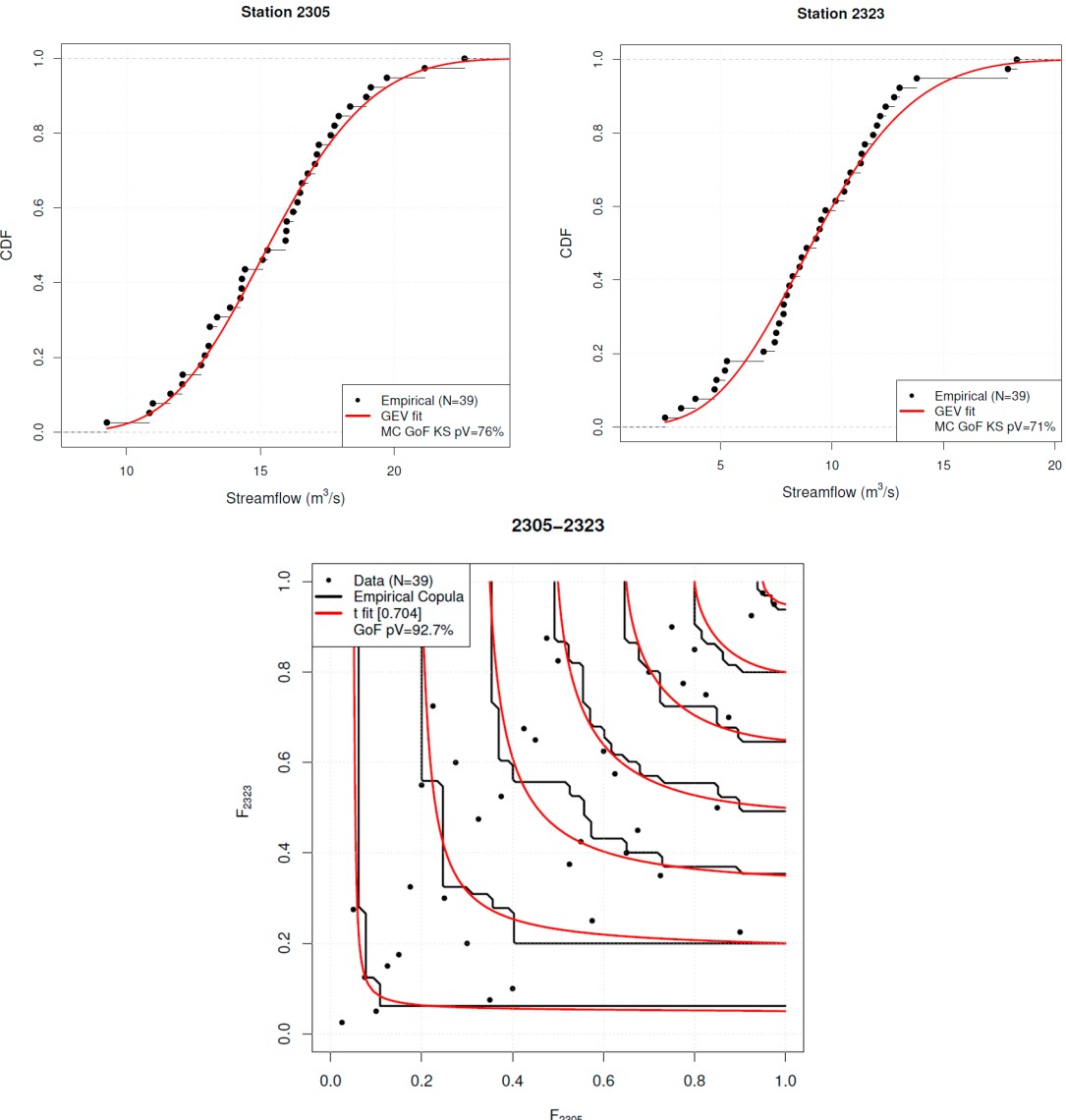

**Figure 6.** Middle Çoruh basin. (**Top row**) GEV fits of the 7-day low flow series in the middle Çoruh basin. Shown: the empirical CDF (black markers), the fitted CDF (red line), the sample size N and an approximate Monte Carlo Goodness-of-Fit test *p*-value of Kolmogorov–Smirnov type. (**Bottom row**) Copula fit. Shown: the pseudo-observations (markers), the empirical copula (black lines) and the fitted copula family (red lines). Additionally shown are the sample size N, an MLE estimate of the copula parameter, and an approximate Monte Carlo Goodness-of-Fit test *p*-value of Cramér-von Mises type.

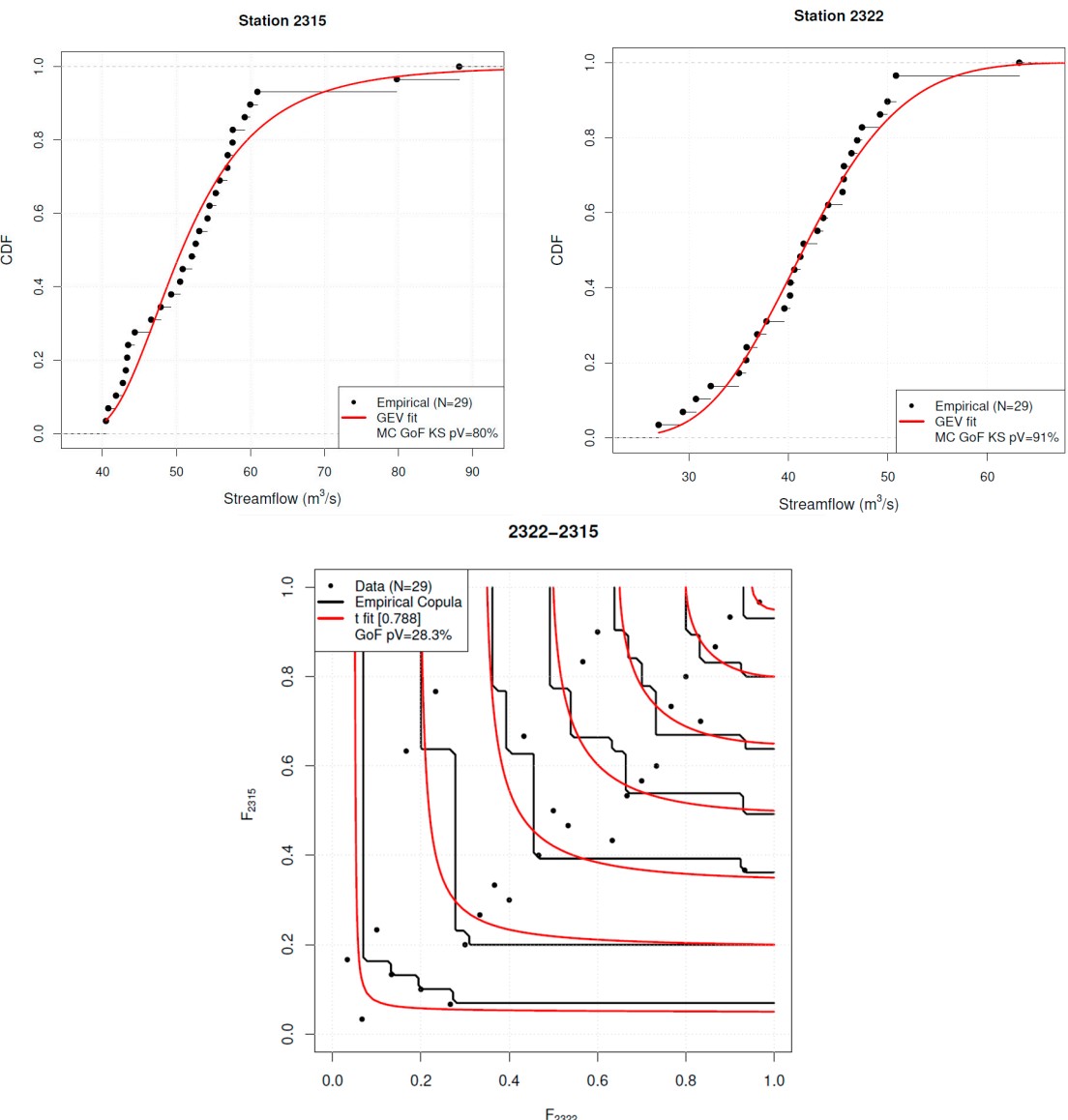

**Figure 7.** Lower Çoruh basin. (**Top row**) GEV fits of the 7-day low flow series in the lower Çoruh basin. Shown: the empirical CDF (black markers), the fitted CDF (red line), the sample size N and an approximate Monte Carlo Goodness-of-Fit test *p*-value of Kolmogorov–Smirnov type. (**Bottom row**) Copula fit. Shown: the pseudo-observations (markers), the empirical copula (black lines) and the fitted copula family (red lines). Additionally shown are the sample size N, an MLE estimate of the copula parameter and an approximate Monte Carlo Goodness-of-Fit test *p*-value of Cramér-von Mises type.

### 3.2. Copula Selection of 7-Day Low-Flow Series

Table 3 shows the estimates of both Pearson and Kendall correlation coefficients, as well as the corresponding *p*-values; in particular, small *p*-values (e.g., less than 5%) indicate that the pairs of AM7 flows considered are not independent. As a result, all the pairs of gauge stations investigated show a clear concordance (i.e., they are not independent); hence, a multivariate (copula) approach is needed in order to correctly model the dependence between the corresponding time series in the upper, middle, and lower parts of the Çoruh basin.

**Table 3.** Estimated Pearson and Kendall correlation coefficients for the pairs of 7-day low-flow series: the values in parentheses are the corresponding *p*-values.

| Coefficient | (2304, 2320) | (2305, 2323) | (2315, 2322) |
|---|---|---|---|
| Pearson | 0.4956 (0.0013) | 0.7510 ($3.65 \times 10^{-8}$) | 0.785 ($4.61 \times 10^{-7}$) |
| Kendall | 0.3457 (0.002) | 0.4700 ($2.69 \times 10^{-5}$) | 0.5894 ($7.99 \times 10^{-6}$) |

Here, 11 families of copulas are considered as candidates for modeling the dependence between the pairs of gauge stations. Several copulas provide admissible and valuable fits of the available data: in turn, the best family is chosen selecting the one associated with the smallest AIC. As a result, a Husler–Reiss copula is used for the upper part of the Çoruh basin, and a t-Student copula for the middle and lower parts (Table 4). Figures 5–7 show the fits of the copulas and provide some statistical information. It is interesting to note that, quantitatively, the parameters of the fitted *t*-copulas in the middle and lower parts of the Çoruh basin are practically the same, indicating that the joint dynamics of the streamflows are very similar at both pairs of stations. In turn, given the fact that copulas rule the joint hazards (see [42]), it will not be a surprise that the plots of the failure probabilities in the middle and lower parts of the Çoruh basin presented later will be matching and comparable.

**Table 4.** Fitted copula families. Shown: the estimates of the parameters and the corresponding variances, the estimates of the corrected AIC, and the Goodness-of-Fit *p*-value.

| Fitted Copulas for the Pair (2304,2320) | | | | |
|---|---|---|---|---|
| **Copula** | **Par. Est.** | **Par. Var.** | **cAIC** | **GoF *p*-value** |
| huslerReissCopula | 1.307 | 0.08289 | −10.54 | 66.58% |
| **Fitted Copulas for the Pair (2305,2323)** | | | | |
| **Copula** | **Par. Est.** | **Par. Var.** | **cAIC** | **GoF *p*-Value** |
| t | 0.7191; 4 | 0.01263 | −25.83 | 91.66% |
| **Fitted Copulas for the Pair (2322,2315)** | | | | |
| **Copula** | **Par. Est.** | **Par. Var.** | **cAIC** | **GoF *p*-Value** |
| t | 0.7987; 4 | 0.006353 | −23.63 | 26.52% |

*3.3. Bivariate Return Periods*

Using Equations (1) and (2), several RPs of interest for the low-flow data in the three Çoruh sub-basins can be computed. The results are shown in Table 5, and the explanation is as follows. For a fixed RP, the univariate design values for each station are computed (the first six left columns in Table 5). Then, the corresponding joint AND RPs of the stations' pairs, (2304,2320)—upper, (2305,2323)—middle and (2315,2322)—lower basin, using the univariate design values as critical thresholds to define the bivariate hazard scenario of interest, are shown in columns 7–9. Evidently, as expected, the relations between the univariate and bivariate RPs previously mentioned well hold; in fact, for a given univariate RP, every pair of design values shows an AND RP larger than the univariate one.

**Table 5.** Design values for given univariate return periods (in years) of the 7-day low-flow series at the gauge stations of interest. Additionally shown are the corresponding bivariate AND return periods estimated at the pairs of station considered in the study—see text.

| RP | Station | | | | | | (2304,2320) Upper | (2305,2323) Middle | (2315,2322) Lower |
|---|---|---|---|---|---|---|---|---|---|
| (years) | 2304 | 2320 | 2305 | 2323 | 2315 | 2322 | $T_{AND}$ | $T_{AND}$ | $T_{AND}$ |
| | (m³/s) | (m³/s) | (m³/s) | (m³/s) | (m³/s) | (m³/s) | (years) | (years) | (years) |
| 5 | 3.68 | 5.39 | 12.94 | 6.34 | 44.94 | 35.39 | 12.2 | 8.4 | 7.6 |
| 10 | 3.42 | 4.63 | 11.78 | 5.03 | 42.67 | 32.49 | 35.9 | 18.7 | 16.5 |
| 20 | 3.21 | 4.00 | 10.88 | 4.02 | 41.08 | 30.25 | 105.4 | 39.8 | 34.8 |
| 50 | 3.00 | 3.28 | 9.90 | 2.93 | 39.52 | 27.86 | 437.9 | 104.5 | 90.4 |

Table 5 provides interesting quantitative information concerning the frequency analysis of the data. On the one hand, it shows the rate at which the streamflow decreases by increasing the return period (the first six left columns). On the other hand, columns 7–9 show the rate at which the joint AND RP increases by considering more and more threatening hazard scenarios (i.e., for decreasing design streamflows). Clearly, these results may help the Water Authority in deciding and planning suitable drought mitigation strategies (such as drought control operating structures and counter-measures), according to the specific (local) water management policies; in fact, based on the univariate information, it would be possible to get some hint on the overall (joint-AND) behavior of low-flow in a whole sub-basin.

Figures 8–10 show the estimates of the univariate low-flow design values associated with different return periods. Additionally shown are Monte Carlo boxplots providing an estimate of the corresponding uncertainty: the whiskers correspond to an approximate confidence interval of approximate 98% level. It is interesting to note that, quantitatively, the uncertainties are small in all cases: practically, of the order of a few m$^3$/s, or less. From a hydrological point of view this means that the fitted model does provide significant information about the design values for different return periods.

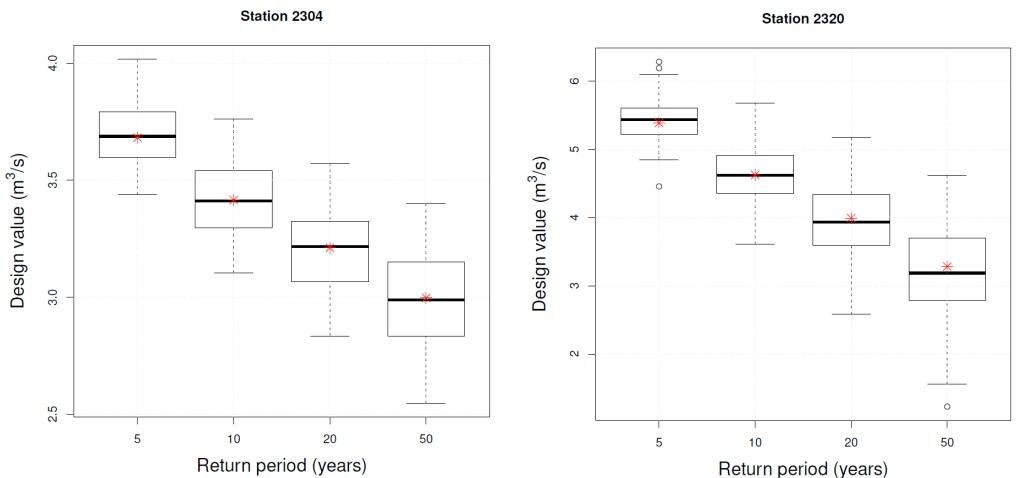

**Figure 8.** Upper Çoruh basin. Estimates of the univariate low-flow design values (red asterisks) associated with different return periods. Additionally shown are Monte Carlo boxplots providing an estimate of the corresponding uncertainty—see text.

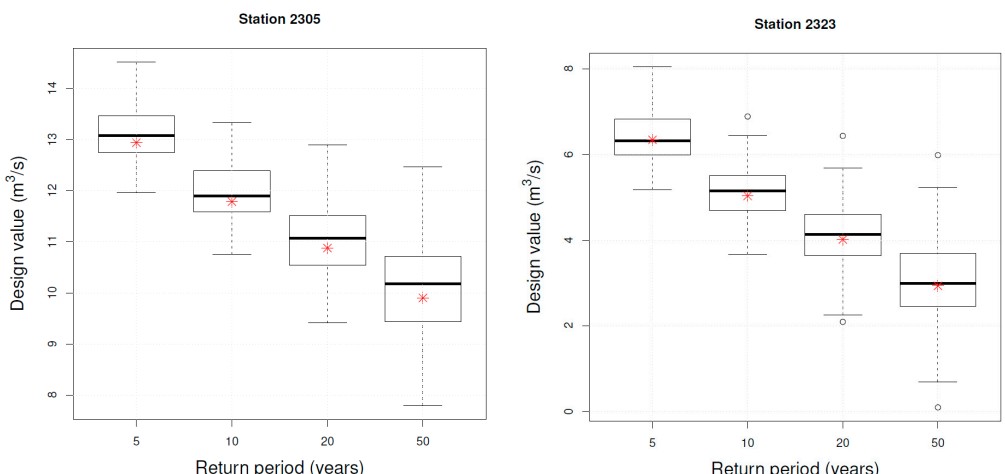

**Figure 9.** Middle Çoruh basin. Estimates of the univariate low-flow design values (red asterisks) associated with different return periods. Additionally shown are Monte Carlo boxplots providing an estimate of the corresponding uncertainty—see text.

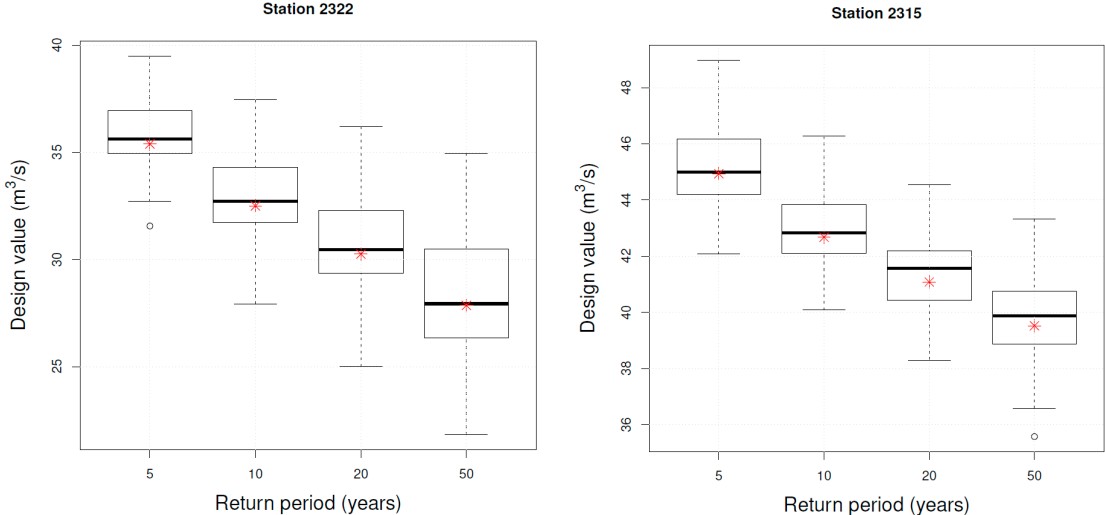

**Figure 10.** Lower Çoruh basin. Estimates of the univariate low-flow design values (red asterisks) associated with different return periods. Additionally shown are Monte Carlo boxplots providing an estimate of the corresponding uncertainty—see text.

### 3.4. Bivariate Failure Probabilities

The notion of failure probability introduced in Section 2.5 is used here in order to assess the hydrological hazard—see [42]), and references therein. Here, the design life time is 50 years. The $p$s in Equation (3) can easily be computed; actually, they are simply the denominator in Equation (2). Four different AND hazard scenarios are considered. These are constructed in such a way that the marginal critical thresholds are given by the quantiles reported in Table 5, corresponding to the four different return periods indicated in the main titles. The results are shown in Figures 11–13.

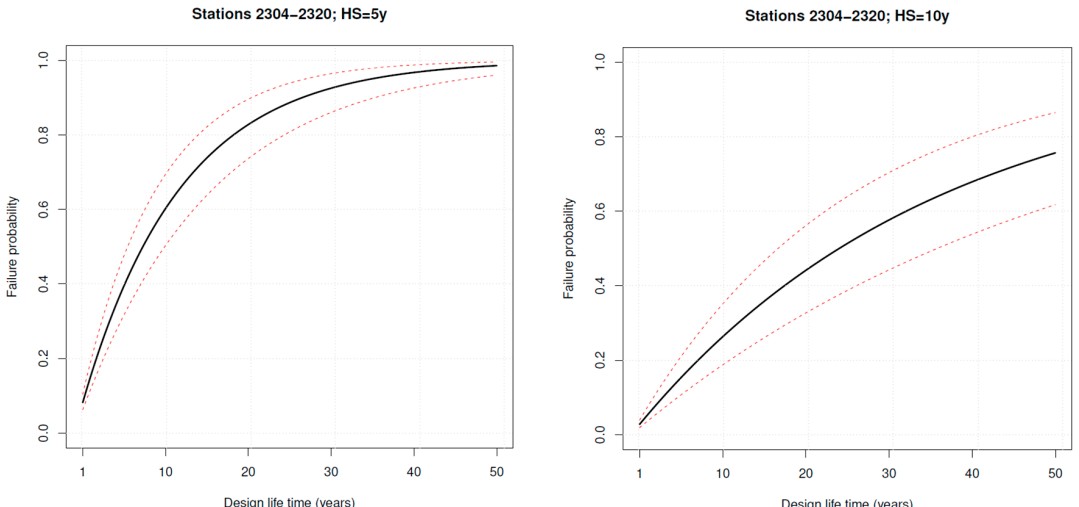

**Figure 11.** *Cont.*

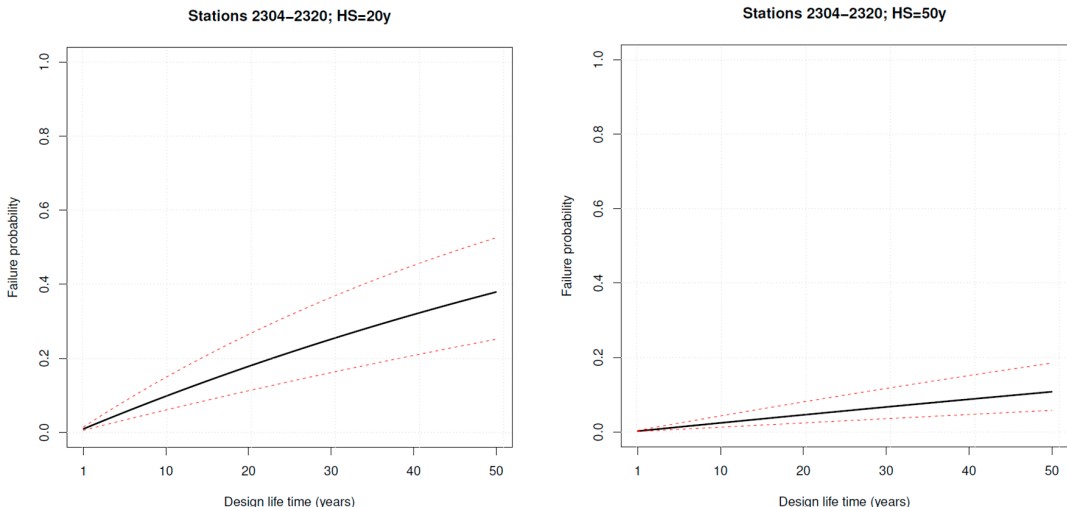

**Figure 11.** Upper Çoruh basin. Failure probabilities for given design life times; the dashed red lines indicate a 95% confidence band. The different plots correspond to the probability that the system enters a bivariate AND hazard scenario whose marginal critical thresholds are given by the quantiles reported in Table 5, corresponding to different return periods indicated in the main titles—see text.

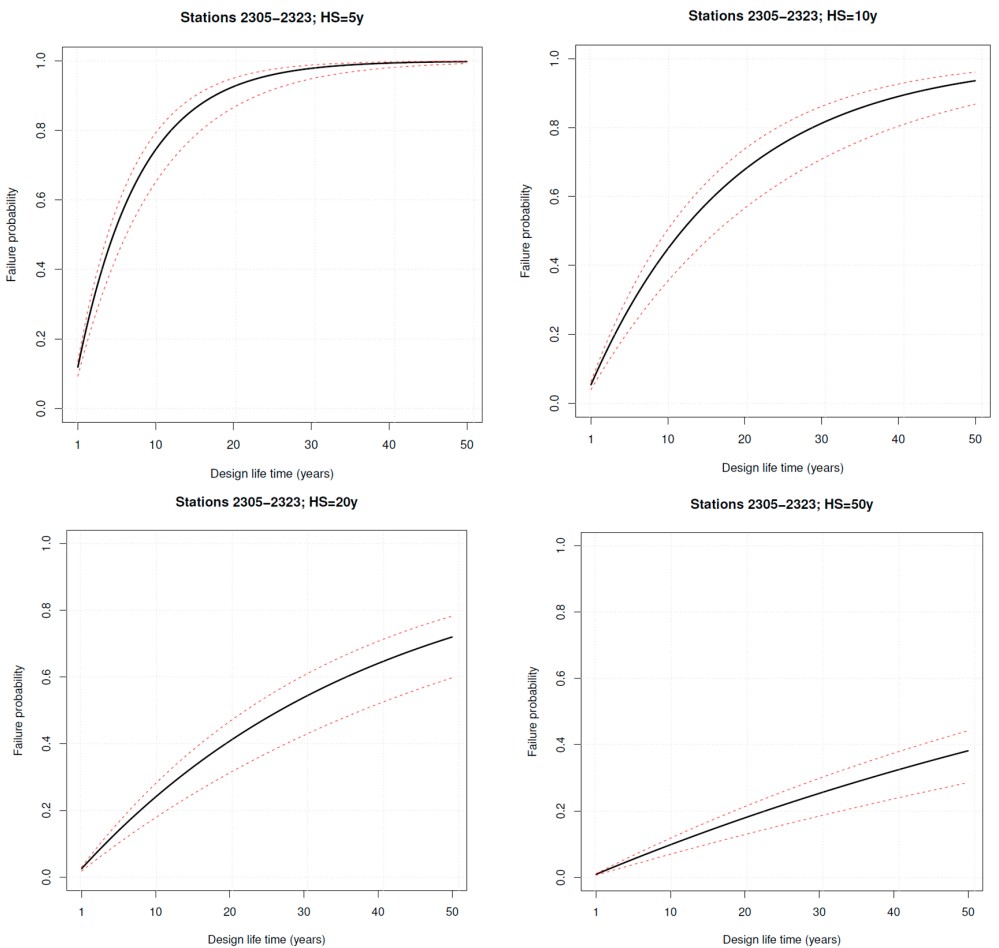

**Figure 12.** Middle Çoruh basin. Failure probabilities for given design life times; the dashed red lines indicate a 95% confidence band. The different plots correspond to the probability that the system enters a bivariate AND hazard scenario whose marginal critical thresholds are given by the quantiles reported in Table 5, corresponding to different return periods indicated in the main titles—see text.

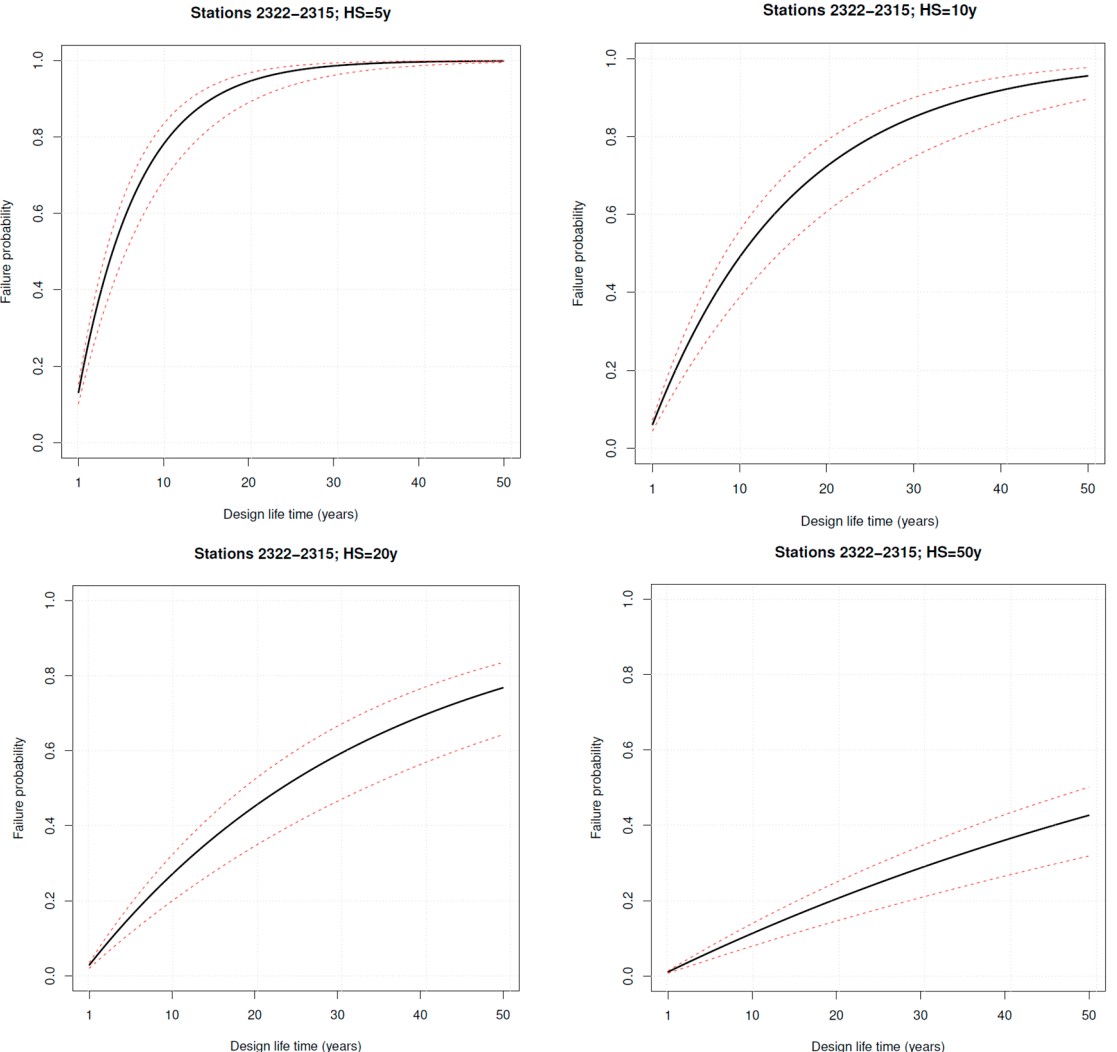

**Figure 13.** Lower Çoruh basin. Failure probabilities for given design life times; the dashed red lines indicate a 95% confidence band. The different plots correspond to the probability that the system enters a bivariate AND hazard scenario whose marginal critical thresholds are given by the quantiles reported in Table 5, corresponding to different return periods indicated in the main titles—see text.

The analysis of the plots yields the following considerations.

- The failure probabilities increase by decreasing the return periods associated with the hazard scenarios in the four panels of each figure. This is obvious since an *HS* corresponding to a small RP (i.e., more frequent) is more likely to be entered by the system than an *HS* associated with a large RP (i.e., more rare).
- The FPs plotted in the figures well characterize the hazards corresponding to different RPs; in fact, the confidence bands practically do not overlap.
- Whatever it is the *HS* considered, the FPs associated with the upper Çoruh basin are smaller than the ones computed for the middle and lower parts (typically, about one half considering HSs of 20 and 50 years). Instead, as already mentioned above, the behavior of the middle and lower sub-basins is quite similar in all cases (this may not be surprising, since the corresponding copulas are quite similar—see Table 4 and Figures 6 and 7). Practically, this means that droughts are less menacing in the upper part of the basin, an important information for the Water Agency.
- The important information extracted from the analysis is that the probability that both the low-flows at paired stations are less than given critical threshold increases from the upper to

the lower part of the basin. The reason is that, in the lower part of the basin, the thresholds (due to the different climate conditions) are larger than in the other parts, and hence it is "easier" (i.e., more probable/hazardous) to experience a (local) drought situation. This provides an essential information for the Water Manager concerning different mitigation policies of the hydrological hazard in different parts of the Çoruh basin.

## 4. Conclusions

The aim of the present study was to use copulas for carrying out a 7-day low-flow analysis in different sub-basins of the Çoruh River Basin (Turkey). Several commonly used univariate distributions, as well as a dozen different copulas, are utilized to model the available low-flow data observed at three pairs of certified gauge stations located, respectively, in the upper-, middle- and lower-part the Çoruh Basin. Suitable univariate marginals and copulas are selected via a thorough statistical analysis. Then, these statistical models are used to compute both return periods and failure probabilities, under a bivariate AND hazard paradigm. Such a multivariate analysis represents a novelty, by combining thorough distributional/frequency/hazard statistical comparative investigations of the streamflow dynamics over all the basin, as outlined and commented in different Sections. Overall, it provides useful indications for the Water Agency in charge of the basin supervision. As a result, the failure probabilities increase from the upper to the lower part of the Çoruh basin. In turn, as pointed out in the work, the upper part of the basin has a smaller drought (joint low-flow) hazard than the middle and lower parts (the hazard is about 50% less considering severe droughts associated with large return periods, like 20 or 50 years).

Since the Çoruh basin will host a huge hydroelectric investment chain, more attention should be paid to the multivariate structure of severe and extreme droughts in this region, in order to effectively manage the water resources. Overall, this study provides useful information about drought occurrences and dynamics in the Çoruh basin, and represents a significant contribution for the Water Agencies taking care of the basin.

**Author Contributions:** Conceptualization, G.S., F.T. and M.Y.; methodology, G.S., F.T. and M.Y.; statistical analysis, and visualization, G.S. and F.T.; data acquisition, F.T. and M.Y.; writing—original draft preparation, G.S., F.T. and M.Y.; writing—review and editing, G.S., F.T. and M.Y; funding acquisition, G.S., F.T. and M.Y. All authors have read and agreed to the published version of the manuscript.

**Funding:** This research received no external funding.

**Acknowledgments:** The authors thank the General Directorate of State Hydraulic Works (Turkey) for providing the streamflow data series. The support of CMCC-Centro Euro-Mediterraneo sui Cambiamenti Climatici-Lecce (Italy) is acknowledged. The support of the European COST Action CA17109 "DAMOCLES" (Understanding and Modeling Compound Climate and Weather Events) is acknowledged. The support of the Italian PRIN 2017 (Research Projects of National Interest) "Stochastic Models of Complex Systems" [2017JFFHSH] is acknowledged.

**Conflicts of Interest:** The authors declare no conflict of interest.

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
