# Peer review of "Multivariate Assessment of Low-Flow Hazards via Copulas: The Case Study of the Çoruh Basin (Turkey)"

_water, doi:10.3390/w12102848_

Round 1
Reviewer 1 Report
See my comments on the file attached.

Reviewer 2 Report
This study utilized 7-day low flows from 6 gauge stations and coupled to three pairs. Then the Authors modeled bivariate probability distribution via copulas.
The submitted paper is well-written and very straight-forward in order; (1) marginal distribution, (2) copula model, (3) joint return period.
However, The Authors need to emphasize the novelty of the paper more because it is obvious that the lower basins are at great risk in drought. In the result and conclusion chapter, there is no quantitative description of the results.
From Line 27 ~ : citation style should be like [1], [2], ...[#]
Lines 150, 234: recommend not to use the word 'much'. The authors don't have to make readers feel like it is not quantitative. Readers will know the Authors intent without that word.
Figures 6, 7, 11, 12, 13: I don't know what happened here but figure labels do not appear appropriately. It is hard to review without those numbers. please check the converted PDF file and figures again.
Lines 341 ~ 358, 360 ~ 373: Recommend to describe results from quantitative perspectives. It is not required to describe all the results the authors made but the submitted paper has no quantitative description at the end of the results and conclusion.
Round 2
Reviewer 2 Report
The paper has been improved a lot.